# Cytotoxicity of β-Cyclodextrins in Retinal Explants for Intravitreal Drug Formulations

**DOI:** 10.3390/molecules26051492

**Published:** 2021-03-09

**Authors:** Manisha Prajapati, Gustav Christensen, François Paquet-Durand, Thorsteinn Loftsson

**Affiliations:** 1Faculty of Pharmaceutical Sciences, University of Iceland, Hofsvallagata 53, IS-107 Reykjavik, Iceland; map52@hi.is; 2Institute for Ophthalmic Research, University of Tübingen, Elfriede-Aulhorn-Strasse 5-7, 72076 Tübingen, Germany; gustav.christensen@uni-tuebingen.de (G.C.); francios.paquet-durand@klinikum.uni-tuebingen.de (F.P.-D.)

**Keywords:** cyclodextrin, retinal explant, cytotoxicity, uptake

## Abstract

Cyclodextrins (CDs) have been widely used as pharmaceutical excipients for formulation purposes for different delivery systems. Recent studies have shown that CDs are able to form complexes with a variety of biomolecules, such as cholesterol. This has subsequently paved the way for the possibility of using CDs as drugs in certain retinal diseases, such as Stargardt disease and retinal artery occlusion, where CDs could absorb cholesterol lumps. However, studies on the retinal toxicity of CDs are limited. The purpose of this study was to examine the retinal toxicity of different beta-(β)CD derivatives and their localization within retinal tissues. To this end, we performed cytotoxicity studies with two different CDs—2-hydroxypropyl-βCD (HPβCD) and randomly methylated β-cyclodextrin (RMβCD)—using wild-type mouse retinal explants, the terminal deoxynucleotidyl transferase dUTP nick end labeling (TUNEL) assay, and fluorescence microscopy. RMβCD was found to be more toxic to retinal explants when compared to HPβCD, which the retina can safely tolerate at levels as high as 10 mM. Additionally, studies conducted with fluorescent forms of the same CDs showed that both CDs can penetrate deep into the inner nuclear layer of the retina, with some uptake by Müller cells. These results suggest that HPβCD is a safer option than RMβCD for retinal drug delivery and may advance the use of CDs in the development of drugs designed for intravitreal administration.

## 1. Introduction

Cyclodextrins (CDs) are cyclic oligosaccharides consisting of (α-1,4-)-linked α-D-glucopyranose units with a hydrophobic central cavity and a hydrophilic outer surface. They can form water-soluble inclusion complexes with numerous lipophilic drugs provided that their structure (or part of it) fits in the CD cavity. No covalent bonds are formed or broken during the complexation and drug molecules in the complex are in rapid equilibria with free molecules in the aqueous complexation media [1]. The complexation affects many physicochemical properties of drugs, such as their aqueous solubility and chemical stability [2]. Natural CDs (i.e., αCD, βCD, and γCD) have a limited solubility in water and, thus, CD derivatives with an improved solubility have been synthesized and are currently used as solubilizing complexing agents in various marketed pharmaceutical products, particularly 2-hydroxypropyl-β-cyclodextrin (HPβCD) and sulfobutylether-β-cyclodextrin (SBEβCD), as well as randomly methylated β-cyclodextrin (RMβCD), although to a lesser extent [1]. CDs have also undergone extensive safety studies and have been approved by both the European Medicines Agency (EMA) and the Food and Drug Administration (FDA) for pharmaceutical use and in dietary supplements [3,4]. CDs have been the subject of numerous review publications [5,6,7,8,9,10,11,12,13].

Drugs for the treatment of retinal diseases are most often delivered via intravitreal injections or implants, where the drug is administered directly into the vitreous humor, which is the hydrogel-type fluid that occupies the space between the lens and the retina. The vitreous mainly consists (99%) of water in a network of collagen and hyaluronic acid. In humans, the volume of the vitreous humor is about 4 mL [14]. Drug molecules must be dissolved in the aqueous vitreous to permeate into the retinal tissue. After an intravitreal injection, hydrophilic and high molecular weight drugs (e.g., proteins and peptides) are known to be excreted via an anterior route to the aqueous humor, while small lipophilic drugs easily pass the retina and are removed via the posterior choroidal flow [15]. The half-life of a dissolved drug in the vitreous humor is typically less than 10 to 24 h, where small molecules have a shorter half-life than biomolecules such as proteins [16,17]. It is expected that the hydrophilic CD molecules (molecular weight between about 1000 and 2000 Da) are readily removed from the vitreous humor after an intravitreal injection. CDs might be able to enhance the retinal delivery of poorly soluble lipophilic drugs after intravitreal administration.

The ability of CDs to complex biomolecules depends upon the molecular structure and the CD binding constants and generally follows the order carbohydrates << nucleic acids << proteins < lipids [18]. Therefore, most biological effects of CDs are based on their interaction with membranes rich in lipids and their ability to extract lipids (e.g., phospholipids and cholesterol) from the plasma membrane [18]. In ophthalmic drug delivery, CDs have mainly been applied topically to the eye, with little or no reports on their intravitreal administration [19,20,21]. Although hydrophilic CDs, such as HPβCD and SBEβCD, have been shown to be well-tolerated when applied topically to the eye, with no detectable side effects, their parent βCD and RMβCD are known to extract cholesterol from cell membranes and form cholesterol/CD complexes [22,23,24]. Furthermore, Nociari et al. observed that βCD extracted lipofuscin bisretinoids from the retinal pigment epithelium (RPE) [25]. Based on these observations, it was proposed that CDs can be used to develop therapeutic candidates for many retinal degenerative diseases, such as Stargardt disease, which is an inherited form of macular degeneration causing central vision loss and sometimes referred to as juvenile macular degeneration. The cause of Stargardt disease is characterized by an abnormal accumulation of lipofuscin in the retina [26]. Similarly, the topical administration of HPβCD as eye drops over 3 months has shown a significant efficacy in reducing amyloid-beta and inflammation in aged mouse retina, consequently improving the retinal function by elevating retinal pigment epithelium-specific protein 65 (RPE 65), which is a key molecule in the visual cycle [27]. Even oral HPβCD treatment in mice resulted in a reduction in the retinal cholesterol content and changes in the retinal sterol, gene, and protein levels [28].

Our study aimed to examine the cytotoxicity of CDs in mouse retinal explants using βCD derivatives to explore their applicability for future ophthalmic formulations. Additionally, fluorophore-conjugation to the same CDs was used to trace the uptake and localization within the retina (Figure 1).

## 2. Results and Discussion

### 2.1. Particle Size and TEM Data Analysis

The particle size of 100 mM CD aggregates was measured by Nano Sight and confirmed by Transmission Electron Microscopy (TEM) (Figure 2). CDs tend to form nano-sized aggregates when their concentration is increased [29]. Although the aggregation in pure aqueous CD solutions is generally low, the aggregation is frequently enhanced by the formation of drug/CD complexes. Furthermore, the aggregation and the size of the aggregates increase with an increasing CD concentration [30].

The concentration of CDs used in pharmaceutical formulations depends on the type of CD used, but due to their favorable toxicological and pharmacological profiles, the CD concentrations can be relatively high [31]. However, the limited solubility of the natural CDs, especially βCD, can limit their concentration [1]. Both HPβCD and RMβCD are very soluble in water (Section 3.1), so we used very high concentrations of these CD derivatives to observe their potential toxicity on retinal explants. Aggregates were observed in aqueous HPβCD and RMβCD solutions. The average diameter of the HPβCD aggregate, as determined by Nano Sight, was 148 nm and that of RMβCD was 305 nm. Larger aggregates (diameter > 100 nm) do not have a spherical shape like smaller ones (<100 nm). Instead, they look like clusters of smaller, spherically-shaped aggregates [32]. The aggregate diameters observed with TEM are smaller than those determined by Nano Sight. This can be explained by the TEM sample preparation, where the aggregate size or structure can change during sample preparation.

Generally, it is thought that only the free drug molecules, which have dissociated from the CD complex, are able to permeate cell membranes [1]. However, recent findings have revealed that CD molecules can enter the cells by endocytosis [33] and this may also be true for drug/CD complexes [34]. However, it appears highly unlikely that the large and hydrophilic CD aggregates can enter cells.

### 2.2. Cytotoxicity of β-Cyclodextrin Derivatives in Retinal Explant Cultures

The terminal deoxynucleotidyl transferase dUTP nick end labeling (TUNEL) assay was employed to quantify the number of dying cells in histological sections from retinal explant cultures incubated with CDs (Figure 3).

The relative cytotoxicity of the CDs was expressed as the percentage of TUNEL positive cells in the respective part of the retinal tissue section. They were counted in both the outer retina (outer nuclear layer, ONL) and inner retina (inner nuclear layer, INL) (Figure 3a). TUNEL positive cells were not counted in the ganglion cell layer (GCL) as most of the cells typically degenerate quickly after explant tissue preparation due to the cutting of the optic nerve.

10 and 100 mM CDs were applied on top of the retinal cultures, i.e., the side closest to the GCL, which is the route the CDs would naturally follow after an intravitreal injection. For 100 mM, the TUNEL positive cell values in the INL were about 0.5%, 5%, and 0.6% for the control, RMβCD, and HPβCD, respectively, while for the ONL, they were about 3%, 12%, and 6% for the control, RMβCD, and HPβCD, respectively (Figure 3b). Therefore, both types of CDs were toxic compared to the control when used in 100 mM concentrations. RMβCD was significantly more toxic compared to HPβCD, both in INL and ONL. HPβCD was predominantly toxic to ONL cells, i.e., where the cell bodies of photoreceptors are located. At the 10 mM concentration, RMβCD still exhibited significant toxicity and killed cells in both ONL and INL, while the number of TUNEL positive cells for HPβCD was similar to the control. This demonstrated that the retina could safely tolerate levels as high as 10 mM HPβCD.

The higher toxicity of RMβCD at both concentrations compared to HPβCD may be due to various reasons. RMβCD is a modified βCD where about two-thirds of the hydroxy groups have been replaced by methoxy groups, while in the case of HPβCD, only a few of the CD-hydroxyl groups have been substituted by 2-hydroxypropyl groups, which makes RMβCD more lipophilic, with a logP value of −6 [12,35,36]. The logP value for HPβCD is about −11. The lipophilicity of the CDs clearly affects other properties, such as the solubilizing capacity, tissue irritating effect, hemolytic activity, and surface activity. The more lipophilic the compound, the easier it penetrates the cell layer, even though the permeability of CDs through biological membranes is negligible, as explained by Loftsson et al. [12], because of the size and hydrophilicity of the CD molecules.

Recently, it has been shown that RMβCD can penetrate the epidermis and dermis of the human skin [37]. In addition, relatively high amounts of HPβCD and dimethyl-β-cyclodextrin were absorbed via the rectum of rats and excreted into the urine, suggesting CD complexes may be absorbable through the rectal mucosa [38]. However, the latest findings regarding the endocytosis of CDs gave a whole new perspective of CDs being able to enter cells [33]. CD molecules can easily form complexes with natural hydrophobic molecules including the cellular components based on the host-guest interaction. Phospholipids are the preferred cellular target for αCD and cholesterol for βCD [39]. Because of this property, they have also been described to induce lysis of cell membranes by removal of membrane components such as cholesterol, phospholipids, and proteins [23,40,41].

Cholesterol is one of the major components of the cell membrane constituting 30% of total lipids and plays an important structural role in membrane stability [42]. Since βCD has an affinity for cholesterol, this CD can induce the release of cellular cholesterol directly affecting the cell and biological barriers [40]. Consequently, the cholesterol content of the membranes can decrease thus affecting the function of the cell membrane and disrupting the barrier function of the cell layers [39]. Additionally, it was found that cholesterol extraction caused the destabilization of tight-junction protein complexes, which are localized in lipid rafts [39]. Methylated β-cyclodextrins tend to interact strongly with lipids [43,44], and there is a correlation between the cytotoxic effect and the cholesterol complexation properties of βCD such that the higher complexation with cholesterol increases the toxicity to the cells [22].

In the retina, cholesterol represents >98% of total sterols [45]. The stronger interaction of CD with the lipids/cholesterol in retinal cells can aid in more vigorous cell membrane destabilization and more cell death. Likewise, there are similar reports on Müller glia cells where the cholesterol status plays an important role. Low cholesterol is harmful to the retinal cells; hence, more cholesterol extraction by RMβCD likely causes more toxicity [45]. This idea was further supported by our phase solubility studies of cholesterol with different β-cyclodextrin derivates (Figure 4).

Cholesterol was solubilized five times more effectively by RMβCD compared to HPβCD (Figure 4). Cholesterol had the highest affinity for the lipophilic RMβCD but had a lower affinity for the very hydrophilic cyclodextrins like HPβCD. Cholesterol solubilization was also affected by the structure of the CD derivative, like the number and position of the methyl groups and the presence of ionic groups [22].

Similar results have been obtained when the toxicity of these CDs was tested on different cell lines. The cytotoxicity of methylated βCDs was found to be very high in intestinal Caco-2 cell cultures, while substitution of the βCD with hydroxyl groups drastically decreased the cytotoxicity [46]. However, in another study, HPβCD presented no cytotoxicity up to 200 mM on the same cell lines [22].

Furthermore, RMβCD possesses surface-active properties [47,48] and it even shows a detergent-like mechanism of lipid solubilization when interacting with lipid vesicles. Here, RMβCD was first adsorbed onto the vesicle surface, which was followed by RMβCD partitioning from the aqueous medium into the phospholipid bilayers forming lipid-RMβCD mixed assemblies and finally the lipid solubilization into micelle like aggregates [49]. The cells in the ONL are photoreceptor cells, and cholesterol is an important component of photoreceptor membranes, relevant for the cells function [50,51]. Hence, photoreceptors might suffer more from the cholesterol extraction capacity of the CDs, something that might be particularly relevant for the higher toxicity observed with RMβCD. Additionally, the cell death in the ONL might be exacerbated by an overall higher sensitivity of photoreceptors, when compared to INL cells.

### 2.3. Fluorescent Microscopy of Fluorescently-Labeled Cyclodextrin Derivates to Study Cellular Uptake in Retinal Cultures

Following this, we investigated the overall distribution of fluorescently-labeled CDs in the retina. To this end, RBITC-HPβCD, FITC-HPβCD, and FITC-RMβCD were used and the fluorescent intensity in the inner and outer retina was quantified (Figure 5) and compared to the control specimen to account for auto-fluorescence coming from the tissue itself. No difference between FITC-labeled CDs was observed for the inner and outer retina, indicating that the HPβCD and RMβCD distribute similarly within the retina. Therefore, the difference in cytotoxicity between the two compounds was likely due to their respective effect on the cells, as discussed above, and not because of differences in the overall tissue distribution.

Some subtle differences between FITC-labeled CDs were observed. For FITC-HPβCD, a string-like structure was seen, spanning across the inner and outer retina. This was not as prominent in cultures where FITC-RMβCD was applied (Figure 5a). This type of staining suggests that the CDs had partly been taken up by Müller glial cells.

When using a rhodamine-labeled CD, we observed an elevated signal in the retina. The signal-to-noise ratio was higher for RBITC-HPβCD than for FITC-HPβCD (Figure 5b,c). This allowed us to detect specific uptake in photoreceptors, which supported the result from the cytotoxicity analysis, where HPβCD mainly killed cells in the outer retina.

However, these results should be taken with a grain of salt as the dye labelling could also have affected cell uptake. First of all, we did not observe the same Müller cell uptake in retinas with RBITC-HPβCD, compared with retinas with FITC-HPβCD. Since the rhodamine molecule is positively charged, the rhodamine-conjugated CDs might bind to negatively charged cell membranes and extracellular matrix elements, possibly facilitating cellular uptake. FITC and RBITC derivatives behave differently and are internalized by different processes. The labelling increases the molecular weight and may alter the properties of the parental CDs, while these still retain high water solubility and cannot cross the cell membrane by passive diffusion. However, there are reports suggesting that endocytosis was observed in fluorescent CDs as well [34]. Müller cells have the capacity to assemble and secrete lipoproteins which can be utilized by photoreceptors or inner retinal neurons, serving as an intraretinal source of cholesterol [52]. This could explain the Müller cell uptake of HPβCD. 

While the fluorescent CDs do not behave exactly as their non-fluorescent form, these studies enhance the understanding of the behavior of labelled CD derivatives at the tissue level [53]. Together with the toxicity analysis, these results may be employed in further the development of CDs as drugs or drug carriers for the treatment of retinal diseases.

## 3. Materials and Methods

### 3.1. Materials

Randomly methylated β-cyclodextrin (RMβCD, also referred to as RAMEB) with a degree of substitution (DS) of 12.6 (molecular weight, MW 1312) was purchased from Wacker Chemie (Munich, Germany), and cholesterol from Sigma Chemical Co. (St. Louis, MO, USA), while 2-hydroxypropyl-β-cyclodextrin (HPβCD) with DS 4.2 (MW 1380) was kindly provided by Janssen Pharmaceutica, Belgium. The solubility of HPβCD in water is >600 mg/mL, while that of RMβCD is >500 mg/mL [1].

6-deoxy-6-[(5/6)-fluoresceinylthioureido]-RMβCD, 6-deoxy-6-[(5/6)-fluoresceinylthioureido]-HPβCD and 6-deoxy-6-[(5/6)-rhodaminylthioureido]-HPβCD were kindly provided by CycloLab, Budapest, Hungary. Milli-Q water and sterile saline solution were used for the preparation of CD solution and all other chemicals were commercially available products (Sigma-Aldrich, Saint Louis, MO, USA) of special reagent grade.

### 3.2. Methods

#### 3.2.1. Animals

The C3H wild-type mouse line was used in all studies [54]. Animals were used irrespective of gender and were housed with free access to food and water under standard white light with 12 h light/dark cycles. They were sacrificed at postnatal day (P) 13 by CO_2_ asphyxiation, followed by cervical dislocation. All procedures were performed in accordance with §4 of the German law on animal protection and approved by the animal protection committee of the University of Tübingen (*Einrichtung für Tierschutz, Tierärztlichen Dienst und Labortierkunde; Registration No. AK02/19M, 3 April 2019*).

#### 3.2.2. Assessment of the Retinal Cytotoxicity of β-Cyclodextrin Derivatives

Culturing of Organotypic Retinal Explant Cultures

To study the cytotoxicity of CDs on the retinal tissue, retinas from mice were isolated for culturing for an extended period of time. The detailed protocol is described elsewhere [55], but will be summarized here. Immediately after animal sacrifice, the eyes were enucleated and incubated for 5 min at room temperature (RT) in R16 serum-free culture medium (Gibco, Carlsbad, CA, USA). To promote the removal of the sclera and choroid, the eyes were transferred to a preheated (37 °C) solution of 0.12% proteinase K (MP Biomedicals, Illkirch-Grafenstaden, France) and incubated for 15 min. Afterwards, the eyes were soaked in 1:4 mixtures of 10% FBS/medium to stop the protease reaction. The eyes were dissected under sterile conditions. The retina with the retinal pigment epithelium (RPE) attached was isolated and cultured on a Transwell membrane (polycarbonate, 0.4 µm pore size, COSTAR, NY) with the RPE side facing down in a 6-well plate. Then, 1 mL complete medium (CM, R16 medium with supplements; detailed under [55]) was added to each well. The explants were allowed to recover from the explantation procedure in a sterile incubator (37 °C, 5% CO_2_) for 48 h. The CM was exchanged every second day by removing 0.7 mL of the CM in the plate and adding 0.9 mL fresh CM to account for evaporation and conserve neuroprotective agents produced by the retinal culture. At P15, CD and saline solution were applied to the cultures for a 48 h incubation period. Here, 20 µL of isotonic CD solutions (adjusted by NaCl) was carefully placed on the top of the retina to cover the whole tissue. Either 10 or 100 mM CD was used. For fluorescently-tagged CD, 5 mM was applied. Alternatively, a 0.9% NaCl solution was added as a control. All solutions were passed through sterile filters (PES, 0.22 µm, Merck Millipore, Ireland) before being introduced to the culture.

Preparation of Retinal Tissue Sections

After the culturing of organotypic retinal explants with CD or saline, the cultures were fixed in a 4% paraformaldehyde/PBS solution for at least 45 min. The explants were cryoprotected by introducing an incremental amount of sucrose to the well plate, i.e., 10% sucrose, 20% sucrose, and 30% sucrose for 10, 20, and 30 min (at RT), respectively. Afterwards, the area of the retina culture attached to the Transwell membrane was cut out with fine scissors, and the membrane piece was submerged in an embedding medium (Tissue-Tek O.C.T. Compound, Sakura Finetek Europe, Netherlands), followed by snap freezing in liquid nitrogen. The frozen specimens were sectioned on an NX50 cryostat (ThermoFisher, Waltham, MA) to produce 14 µm thick sections on Superfrost Plus object slides (R. Langenbrinck, Emmendingen, Germany) used for direct imaging or further staining.

Assessing Cell Death in Retinal Sections Using the TUNEL Assay

A terminal deoxynucleotidyl transferase dUTP nick end labeling (TUNEL) assay was used to stain the nuclei with damaged (nick-end) DNA [56]. This was done to quantify the number of dying cells in retinal cultures after CD or saline was applied to assess the cytotoxicity of the CDs. Firstly, microscopy slides with retinal sections were rehydrated with PBS. A proteinase K solution preheated to 37 °C (0.21 µg/mL, Tris-buffered saline (TBS) (Sigma-Aldrich, Saint Louis, MO, USA) was added and incubated for 5 min at 37 °C. After washing with TBS, a solution of ethanol/acetic acid was added and incubated for 5 min before washing. A blocking solution consisting of 1% bovine serum albumin, 10% normal goat serum, 3% Triton-X (Sigma-Aldrich, Saint Louis, MO, USA), and 2.5% fish gelatin in PBS was incubated on the sections (1 h, RT). A TUNEL reaction solution consisting of the enzyme solution and labeling solution from the In Situ Cell Death Detection Kit (TMR red, Product No. 12156792910, Sigma Aldrich) was prepared at a 1:9 ratio, diluted in blocking solution (1:1), and incubated with the slides at 37 °C for 1 h. The slides were washed with PBS, and a mounting medium with DAPI (Vectashield, Vector Laboratories, USA) was added. Samples were kept at 2–8 °C for at least 30 min before imaging with fluorescent microscopy (Axio Imager Z2 ApoTome, Carl Zeiss Microscopy GmbH), using a CCD camera with a 20× objective. Ex./Em. of 548/561 nm was used to detect TUNEL labeling at random locations of the section. Image acquisition was conducted by recording z-stacks, each with 10 images 1 µm apart. Values such as the exposure time for the red (TUNEL) channel, binning, and brightness/contrast of each image were kept consistent. To quantify the number of dying cells, in either the inner or outer nuclear layer in the tissue section, the following equation was used:(1)$$\mathrm{TUNEL}\;\mathrm{positive}\;\mathrm{cells}\;\left(\%\right)\;=\;\frac{\#\mathrm{TUNEL}\;\mathrm{positive}\;\mathrm{nuclei}}{\mathrm{Area}\;\mathrm{of}\;\mathrm{layer}\;/\;\mathrm{Average}\;\mathrm{area}\;\mathrm{of}\;\mathrm{nuclei}}\cdot 100\%.$$

To analyze the statistical significance within the dataset, one-way ANOVA with Tukey’s multiple comparisons test (α = 0.05) was performed using GraphPad Prism 8.

Determining the Retinal Uptake of Fluorescently-Labeled β-Cyclodextrin

Fluorescently-labeled CD (FITC-HPβCD, RBITC-HPβCD, and FITC-RMβCD) was added to organotypic retinal explant cultures, as described previously. Slides with retinal tissue sections were rehydrated in PBS for 10 min before mounting medium with DAPI (Vectashield, Vector Laboratories, USA) was added. Images were recorded with fluorescent microscopy, as described above. A green (Ex./Em. 493/513) and red (Ex./Em. 558/575) channel were used to detect the fluorescein or rhodamine labeling, respectively. Image acquisition was conducted by recording z-stacks, each with 14 images 1 µm apart. Values such as the exposure time for the green and red channel, binning, and brightness/contrast of each image were kept consistent. Sections from the saline-treated retinas were imaged using the same parameters to determine the level of green and red auto-fluorescence from the cultures. Alternatively, tile pictures showing the entire retinal section were recorded by stitching together adjacent projected z-stacks in the image acquisition software (ZEN 2.6, Carl Zeiss Microscopy GmbH).

#### 3.2.3. Particle Size Measurement

Nano Sight Wave

The laser-based light scattering analysis of CD particles was performed with Nano Sight NS300 (Malvern, Worcestershire, UK), fitted with an O-ring top-plate. Nanoparticle tracking analysis (NTA) software was used to capture images and process data, representing the concentration, size distribution, and intensity of particles in the sample. Sample measurement was done in static mode using a capture time of 60 s and five repeats. The camera level was adjusted to 11 so that all particles were visible. The same camera level was used for all the samples. A suitable detection level was selected for data analysis to limit the detection of non-particles and was between levels 4 and 12. The result for each sample was based on the average of five measurements obtained from the NTA and represented by the average particle concentration, average particle size (i.e., mean size), and mode size (i.e., the size that displays the highest peak). 

Transmission Electron Microscopy (TEM) Analysis

The size and morphology of the CD in an aqueous solution were evaluated by using Model JEM-1400 Transmission Electron Microscopy (JEOL, Tokyo, Japan). The negative staining technique was utilized. Firstly, a small amount of clear CD solution (3 µL) was dropped on a 300-mesh coated grid and dried at 37–40 °C for one hour. Then, a drop of centrifuged 4% *w*/*v* uranyl acetate (26 µL) was added to the loaded grid. After 6 min of staining, the sample was dried overnight at room temperature. Finally, the stained grid was placed in the sample holder and inserted into the microscope.

#### 3.2.4. Phase-Solubility Studies

The phase-solubility studies of cholesterol were performed by dissolving cholesterol (20 mg/mL) in methylene chloride, followed by solvent evaporation (0.3 mL per vial) under a flow of nitrogen in cylindrical vials. This left a very thin layer of cholesterol on the inner surface of the vials. Aqueous CD solutions were added to the vials, which were tightly sealed and heated in an autoclave (121 °C for 20 min) [57]. The vials were opened after equilibrating them at room temperature overnight and a small amount of solid cholesterol was added to each vial, which were again allowed to equilibrate at room temperature for an additional 6 days. The seeding with cholesterol was performed to reduce the possibility of complications caused by differences in the solid-state of cholesterol. Finally, the aqueous cholesterol suspensions were filtered through a 0.45 µm nylon membrane filter, and the filtrate was analyzed using high-pressure liquid chromatography (HPLC).

Quantitative determination of cholesterol was performed on a reversed-phase HPLC system from Merck-Hitachi (Darmstadt, Germany) consisting of an L 4250 UV-Vis detector operated at 203 nm, L 6200 A Intelligent pump, AS-2000A Autosampler, D-2500 Cromato-Integrator, and Phenomex Luna (Cheshire, UK) 5 µm C18 reversed-phase column (150 × 4.6 mm). The mobile phase consisted of methanol, acetonitrile, isopropyl alcohol, and tetrahydrofuran (50:25:25:0.1).

## Figures and Tables

**Figure 1 molecules-26-01492-f001:**
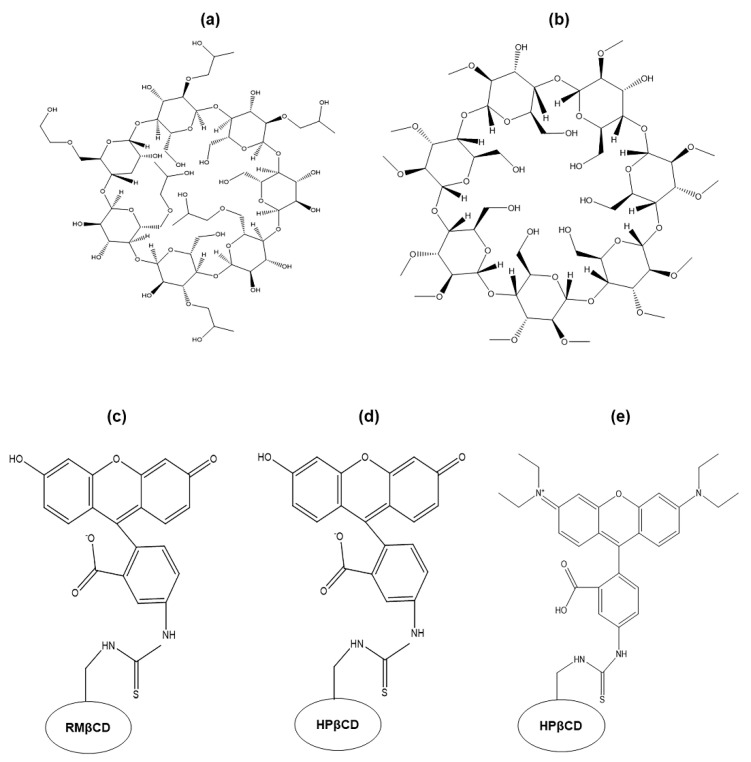
Chemical structures of (**a**) 2-hydroxypropyl-β-cyclodextrin (HPβCD), (**b**) randomly methylated β-cyclodextrin (RMβCD), (**c**) 6-deoxy-6-[(5/6)-fluoresceinylthioureido]-HPβCD (FITC-RMβCD), (**d**) 6-deoxy-6-[(5/6)-fluoresceinylthioureido]-RMβCD(FITC-HPβCD), and (**e**) 6-deoxy-6-[(5/6)-rhodaminylthioureido]-HPβCD (RBITC-HPβCD).

**Figure 2 molecules-26-01492-f002:**
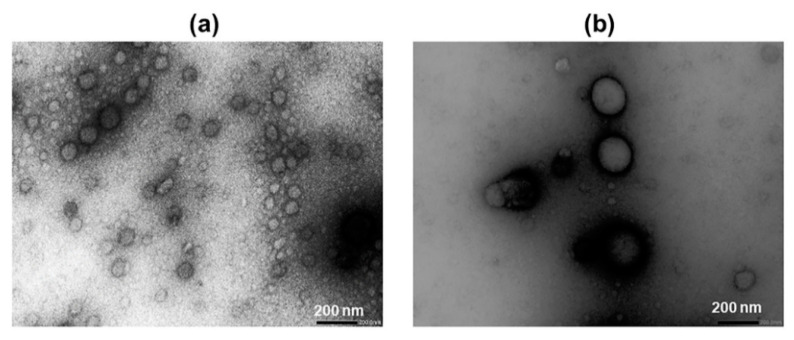
Transmission electron microscopy images of cyclodextrin (CD) aggregates at a magnitude of 30 K. (**a**) 100 mM HPβCD and (**b**) 100 mM RMβCD. The average diameter of RMβCD aggregates appeared to be around two times larger than that of HPβCD.

**Figure 3 molecules-26-01492-f003:**
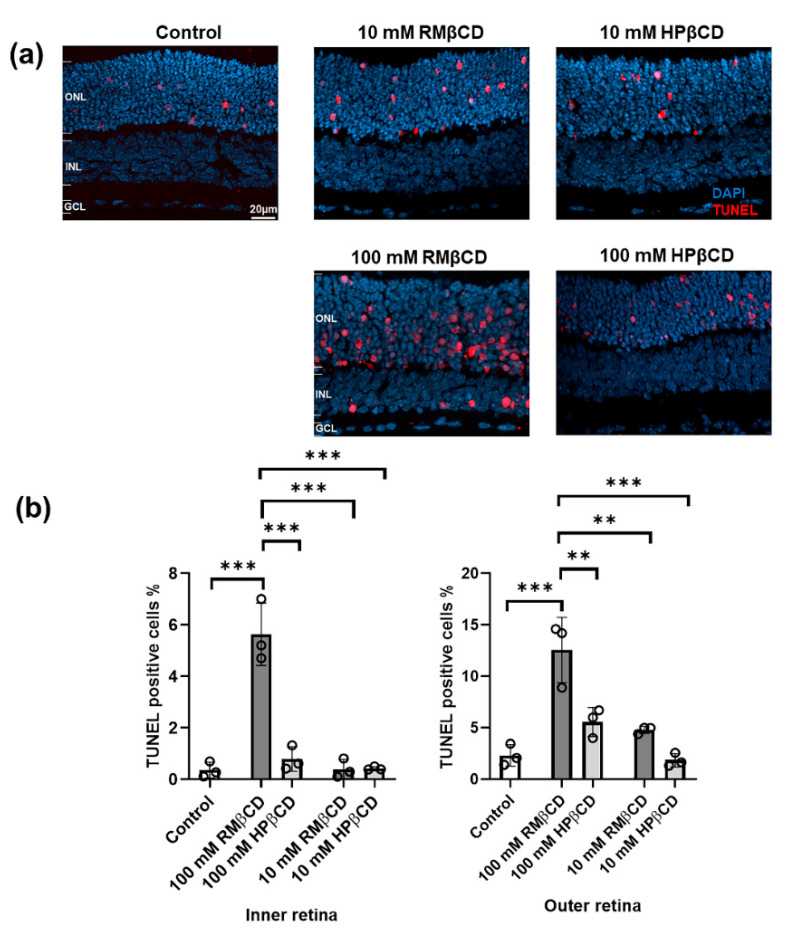
Cytotoxicity of β-derivatives of cyclodextrin in retina explant cultures. (**a**) Sections of retinal cultures to which different CD solutions were applied; saline solution was used as the control. The terminal deoxynucleotidyl transferase dUTP nick end labeling (TUNEL) assay (red) was used to detect dying cells. DAPI (blue) was used as a nuclear counterstain. Cultures were derived from wild-type mice at postnatal day (P) 13. CDs were added at P15 and incubated with the cultures for a duration of 48 h. ONL = outer nuclear layer, INL = inner nuclear layer, and GCL = ganglion cell layer. (**b**) Analysis of average TUNEL positive cells (%) in both INL and ONL from cultures with different CD solutions and the control. Results represent the mean ± SD for *n* = 3 explant cultures per group. Statistical analysis was performed using one-way ANOVA with Tukey’s multiple comparisons test (α = 0.05) and asterisks represent the significant difference (** = *p* ≤ 0.01 and *** = *p* ≤ 0.001).

**Figure 4 molecules-26-01492-f004:**
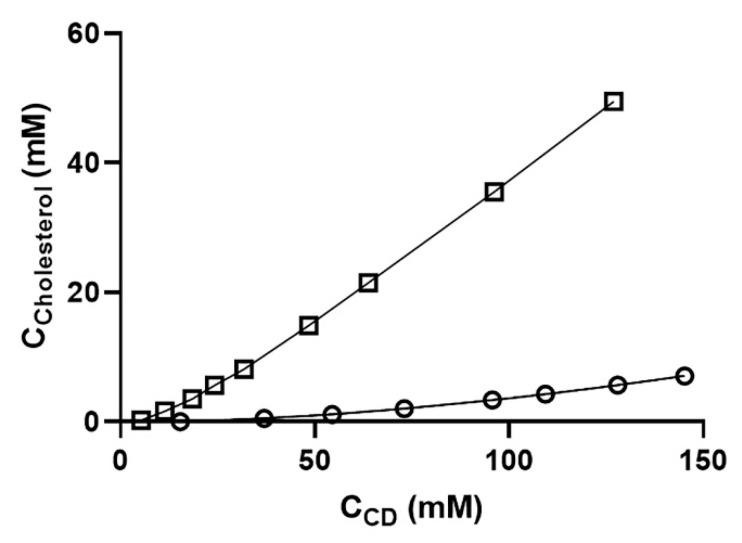
Phase-solubility diagrams of cholesterol in various aqueous β-cyclodextrin derivatives at room temperature. The diagram shows the CD concentration plotted against the cholesterol concentration. Overall, cholesterol was solubilized about five-fold more by RMβCD than by HPβCD. Each point represents the mean of triplicate experiments. Key: ο = HPβCD and **□** = RMβCD.

**Figure 5 molecules-26-01492-f005:**
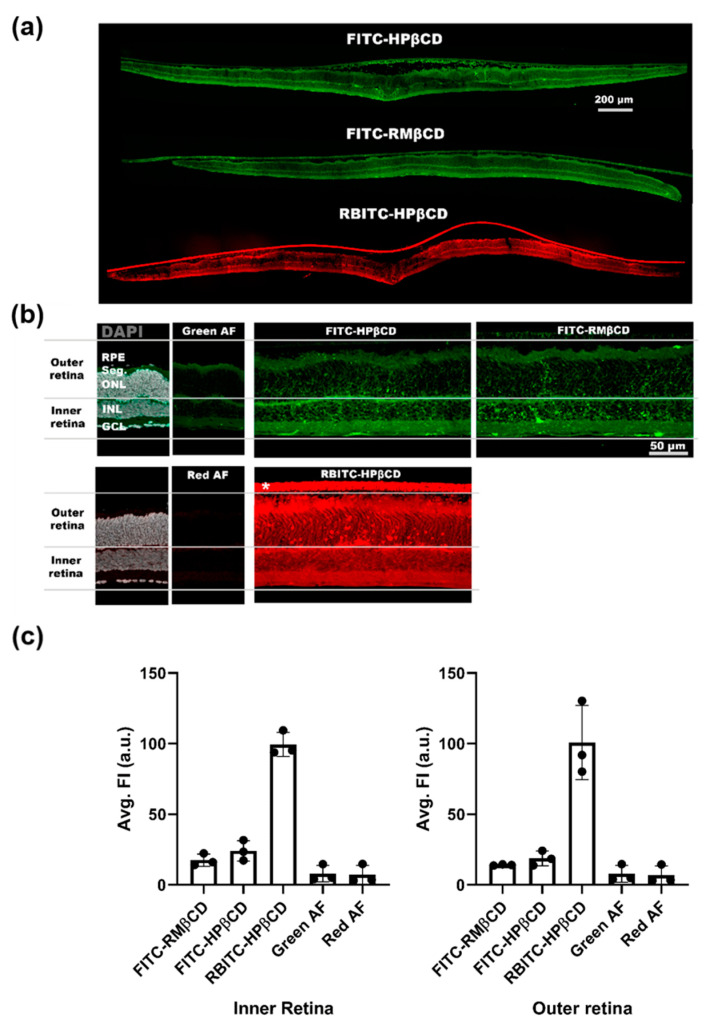
Uptake of fluorescently-labeled CDs in the retina explant culture. (**a**) Overview of tissue sections from wild type (WT) mice explant cultures to which three different fluorescently-labeled CDs were added to the side facing down in the image. (**b**) Close-up images of the sections were taken for an analysis of the fluorescent signal. Images from control sections (without added CDs) were used to measure the intensity of the red and green auto-fluorescence (AF) coming from the tissue itself. For this analysis, the outer retina is defined as the area from the outer nuclear layer (ONL) to the retinal pigment epithelium (RPE), while the inner retina encompasses the area from the ganglion cell layer (GCL) to the inner nuclear layer (INL). Seg. = segments of the photoreceptors (inner and outer segment). * membrane on which the retinal explant was cultured. (**c**) Analysis of the average fluorescent intensity coming from either the inner or outer retina. For both inner and outer retina, a high signal was detected from RBITC-labeled CDs, while the signal was lower and similar for FITC-labeled CDs. Results represent the mean ± SD for *n* = 3, where *n* is the number of animals.

## Data Availability

No data reported.

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
