# Peer review of "Cytotoxicity of β-Cyclodextrins in Retinal Explants for Intravitreal Drug Formulations"

_molecules, 2021, doi:10.3390/molecules26051492_

Round 1

Reviewer 1 Report

Comments

Manuscript Number: molecules-1130302

Title:  Cytotoxicity of Cyclodextrins in Retinal Explants for Intravi-2 treal Drug Formulations

Authors:   Manisha Prajapati, Gustav Christensen, François Paquet-Durand and Thorsteinn Loftsson

The results are very useful and should be published, allowing us to reach a better understanding relative to the cytotoxicity of cyclodextrins in retinal explants for intravitreal drug formulations. The whole work is well organized, with an interesting experimental part, together with appropriate tables and figures.

I consider that only some points must be re-considered by the authors and, consequently, my opinion it is that manuscript needs a minor review before to be published.

The specific comments are shown below.

  1. In page 4, when the authors say:

  1. “…In this study, very high concentrations of CDs were used to observe their potential 105 toxicity on retinal explants and hence aggregates were observed in the aqueous HPβCD 106 and RMβCD solutions…”

 they are invited to clarify this sentence. On the other words, they are invited to insert in this sentence the information concerning the range of concentrations of CDs used depends of the CDs type. In addition, it would be also recommendable that the authors refer in the Introduction or in Materials section the solubility of cyclodextrins used in this work, together with some references, once the solutions of CDs depend of this property. For example, the solubility of b-CD is lower than HP- bCD.   2.      In Figure 4, the authors must write CCD/(mM) instead of CD conc.(mM), IUPAC Rules). Having in mind these points, I recommend reviewing all the text. 

  1. The meaning of symbols in all the manuscript (text and figures) should be indicated.
  2. In Introduction, on page 2, when the authors say:

. The vitreous consists 48 mainly (99%) of water in a network of collagen and hyaluronic acid. In humans, the vol-49 ume of the vitreous humor is about 4 ml [14]. Drug molecules must be dissolved in the 50 aqueous vitreous to be able to permeate into the retinal tissue…”

they identify hyaluronic acid as a component in vitreous. It is known that the hyaluronic acid HA is the major macromolecular component of the intercellular matrix of most connective tissues, such as cartilage, eye vitreous humour, and synovial fluid.

It can be used as drug as well as a carrier in drug delivery systems. So, it is of great interest for many technical fields such as biomedical and pharmaceutical applications.

Thus, my question is: Did the authors ever think of using this carbohydrate as a carrier in these particular controlled-release systems, using as drugs those applied in the treatment of retinal diseases? Another open possibility would be to study the behavior of a combined system between hyaluronic acid and some cyclodextrin.

I congratulate the authors for such a good paper. In conclusion, this is a good manuscript and I think that after minor review, it should be published in this Journal.

Reviewer 2 Report

The manuscript Titled "Cytotoxicity of cyclodextrins in retinal explants for intravitreal drug formulations" is interesting, overall experiments are well planned and appropriate techniques have been used. The results are clearly presented and the manuscript is well written. However, I have a few comments, questions and suggestions.

1) The title is a bit misleading since the authors only have done cytotoxicity study of Beta CD. There is no results on the use of the studied cyclodextrin derivatives for intravitreal drug formulations.

2) Figure 2 (b): The tow graphs are identical. Is it really correct?

3) What is the actual temperature of "Ambient temperature"?

4) Figure 5 and related section: Why author did not choose to study on fluorescent labelling with Rhodamine of RMbetaCD? Why Rhodamine labelling for only HPbetaCD?
